# Adenomatous Polyposis Coli loss controls cell cycle regulators and response to paclitaxel in MDA-MB-157 metaplastic breast cancer cells

Emily M. Astarita[1,2], Sara M. Maloney[1,3], Camden A. Spring[1,4], Bronwyn J. Berkeley[1¤a], Monica K. VanKlompenberg[1,3¤b], T. Murlidharan Nair[5], Jenifer R. Prosperi[1,3,4]*

1 Harper Cancer Research Institute, South Bend, IN, United States of America, 2 Department of Chemistry/Biochemistry, University of Notre Dame, Notre Dame, IN, United States of America, 3 Department of Biochemistry and Molecular Biology, Indiana University School of Medicine, South Bend, South Bend, IN, United States of America, 4 Department of Biological Sciences, University of Notre Dame, Notre Dame, IN, United States of America, 5 Department of Biology and Computer Science/Informatics, Indiana University South Bend, South Bend, IN, United States of America

¤a Current address: BHF/University Centre for Cardiovascular Science, University of Edinburgh, Edinburgh, United Kingdom
¤b Current address: Department of Animal and Avian Sciences, University of Maryland, College Park, MD, United States of America
* jrprospe@iupui.edu

## Abstract

Adenomatous Polyposis Coli (APC) is lost in approximately 70% of sporadic breast cancers, with an inclination towards triple negative breast cancer (TNBC). TNBC is treated with traditional chemotherapy, such as paclitaxel (PTX); however, tumors often develop drug resistance. We previously created APC knockdown cells (APC shRNA1) using the human TNBC cells, MDA-MB-157, and showed that APC loss induces PTX resistance. To understand the mechanisms behind APC-mediated PTX response, we performed cell cycle analysis and analyzed cell cycle related proteins. Cell cycle analysis indicated increased G2/M population in both PTX-treated APC shRNA1 and parental cells, suggesting that APC expression does not alter PTX-induced G2/M arrest. We further studied the subcellular localization of the G2/M transition proteins, cyclin B1 and CDK1. The APC shRNA1 cells had increased CDK1, which was preferentially localized to the cytoplasm, and increased baseline CDK6. RNA-sequencing was performed to gain a global understanding of changes downstream of APC loss and identified a broad mis-regulation of cell cycle-related genes in APC shRNA1 cells. Our studies are the first to show an interaction between APC and taxane response in breast cancer. The implications include designing combination therapy to re-sensitize APC-mutant breast cancers to taxanes using the specific cell cycle alterations.

## Introduction

Breast cancer accounts for 30% of all new cancer diagnoses in women and is the leading cause of death for women between the ages of 20 and 59 years. Compared to other cancers, the

**Data availability statement:** All relevant data are within the manuscript and its Supporting Information files.

**Funding:** This research was supported by the American Cancer Society – Institutional Research Grant, Navari Family Foundation, the Indiana CTSI, grant #UL1 TR001108 from the NIH, NCATS, and a CTSI Core Usage grant (JRP). This work was also supported by an award from the Ralph W. and Grace M. Showalter Research Trust and the Indiana University School of Medicine (JRP). The content is solely the responsibility of the authors and does not necessarily represent the official views of the Showalter Research Trust or the Indiana University School of Medicine. EMA was supported by the University of Notre Dame College of Science. BB was supported by a Naughton Fellowship through the University of Notre Dame. The funders had no role in study design, data collection and analysis, decision to publish, or preparation of the manuscript.

**Competing interests:** The authors have declared that no competing interests exist.

genetic variability that is seen in breast cancer complicates treatment. There exist different subtypes of breast cancer tumors, characterized by their expression of functional receptors. The major subtypes of breast cancer include luminal A, luminal B, HER2 enriched and triple negative. Both luminal A and B are estrogen receptor (ER) positive, while HER2 enriched is human epidermal growth factor 2 receptor (HER2) positive. These hormone responsive subtypes can be treated with more specific therapies. Triple negative breast cancer (TNBC) is a subtype that is named by its lack of all three functioning receptors that are up-regulated in other subtypes: HER2, ER, and progesterone receptor (PR). For this reason, TNBC patients have limited treatment options as the lack of receptors disallows the use of more targeted therapies. As a result, they are treated with traditional chemotherapies, such as taxanes, platinums, and anthracyclines. Despite initial positive response, TNBCs often develop resistance and/or tumor relapse. Paclitaxel (PTX) is a taxane used for TNBC, which is well established to induce mitotic arrest due to activation of the spindle assembly checkpoint [1]. PTX arrests cells through interfering with the treadmilling function of microtubules (MTs), necessary for chromosome segregation and cell division, and inducing changes in cell cycle proteins that are important for mitotic function. While the mechanism of action of PTX is well characterized, PTX resistance remains less explicated.

Adenomatous Polyposis Coli (APC) is a multi-domain protein and acts as a negative regulator to the Wnt/β-catenin signaling pathway to control gene expression, cell proliferation, and differentiation [2]. APC also functions independently of β-catenin in the regulation of MT stability, DNA repair, and cytoskeletal organization through mechanisms that are not fully understood [3]. *APC* is mutated or hypermethylated in up to 70% of sporadic breast tumors with an inclination towards TNBCs [4]. In fact, breast cancers lacking just two of the three hormone receptors see a significant increase in APC methylation compared to tumors with functioning hormone receptors [5].

Findings have demonstrated that APC loss contributes to chemotherapy resistance through disruption of mitotic spindle, enhancing DNA repair, and alterations in expression and activity of ATP-dependent binding cassette transporters [6–8]. In an *in vivo* model of APC loss, cells become less responsive to Taxol treatments. Fewer APC mutant cells went into mitotic arrest after Taxol treatment compared to the wild type cells, and Taxol failed to induce apoptosis in APC deficient cells [6]. In addition, our lab previously showed that APC loss increases PTX resistance in the human MDA-MB-157 metaplastic TNBC cell line [7]. This cell line was specifically chosen due to the formation of squamous metaplasia in $Apc^{Min/+}$ mice, and the resemblance of mammary tumors from the MMTV-PyMT;$Apc^{Min/+}$ mice to the metaplastic subtype of TNBC [9, 10]. Due to the increased incidence of chemotherapeutic resistance in tumors with mutated APC, it is essential to study the uninvestigated roles of APC and gain insight on its implications in treatment efficacy.

The less-commonly investigated roles of APC indicate that it may have significant implications on the efficacy of breast cancer treatments. Therefore, we hypothesized that APC loss may lead to a difference in cell cycle protein modulators during G2/M transition. In this paper we show that the PTX-induced G2/M arrest occurs in both APC shRNA1 and parental cells. In addition, there is a significant increase in total CDK1 and CDK6 expression in the APC shRNA1 cells. PTX treatment conferred downregulation of p27 in MDA-MB-157 and APC shRNA1 cells. RNA-sequencing showed gene expression differences in cell cycle-related transcripts between the parental and APC shRNA1 cells. GLI1, NUPR1, and LBH were confirmed to be up-regulated, while RGS4 was decreased, in untreated APC shRNA1 cells compared to untreated parental cells. These four genes play different roles in cell cycle control and expression of cell cycle proteins, but the combined expression pattern change observed with APC knockdown suggests a connection between APC and regulation of the cell cycle.

Taken together, the observation that APC controls expression of multiple cell cycle genes and proteins is important to understand the mechanism of resistance seen in APC shRNA1 cells, and provides multiple viable targets for combination therapy in PTX-resistant TNBCs.

## Materials and methods

### Cell culture and drug treatment

MDA-MB-157 breast cancer cells (ATCC, Manassas, VA) were maintained at 37°C with 5% $CO_2$ in RPMI 1640 media with 1:5000 plasmocin, 1% penicillin/streptomycin, 10% fetal bovine serum and 25 mM HEPES. The MDA-MB-157 cells were authenticated using STR DNA profiling (Genetica DNA Laboratories, Burlington, NC). Lentiviral mediated shRNA knockdown of *APC* was performed in MDA-MB-157 cells to create the *APC* knockdown shRNA 1 and 2 cell lines (S1 Fig) [7]. The *APC* knockdown shRNA 1 and 2 cell lines were routinely maintained in media containing 1.5 μg/mL puromycin (MilliporeSigma, St Louis, MO). Cells were regularly passaged using 0.25% trypsin/EDTA. Reporter assays were performed as previously described [11]. Briefly, cells were plated in triplicate and transfected using Lipofectamine 2000 (Invitrogen) with either pTOPflash or pFOPflash as previously described, and co-transfected with pRL-TK (Renilla luciferase-thymidine kinase; Promega, Madison, WI). Lysates were analyzed using the Dual Luciferase Assay System kit (Promega). Luciferase activity was normalized for transfection efficiency and FOPflash activity. For drug treatments, cell lines treated at 50–70% confluence with 1 μM nocodazole, 0.078 μM paclitaxel, 16 μM cisplatin, or control DMSO for 24 hours. Nocodazole was used as a control for G2/M arrest, and cisplatin was used as a control because it effectively kills APC shRNA1 cells [7]. All drugs were purchased from MilliporeSigma.

### Flow cytometry

MDA-MB-157 and APC shRNA1 cells were used for each treatment: nocodazole, PTX, or control. After 24 hours of treatment, $2x10^6$ cells were fixed using dropwise addition of 70% ethanol, with vortexing between additions. Cells were fixed for 30 minutes at 4°C and then washed with PBS. Cells were centrifuged and the pellets were resuspended in RNase1 and Propidium Iodide (PI). After PI staining was performed, forward scatter and side scatter were obtained, as well as PI fluorescence, on a Cytotomics FC 500 (Beckman Coulter, Brea, CA) flow cytometer. The percentage of cells in each phase of the cell cycle was determined using FlowJo™ Flow Cytometry Data Analysis Software (Tree Star, Ashland, OR) [12], using GraphPad Prism for statistical analysis. Graph shows average with error bars to illustrate standard deviation. Statistical significance was determined using one-way ANOVA with a post-hoc Sidak test.

To confirm our previous findings of Annexin/PI staining was performed using the AlexaFluor 488 AnnexinV/Dead Cell Apoptosis Kit (ThermoFisher) following the manufacturer's instructions. Briefly, cells are treated with PTX or control as described above for 24 hrs, followed by staining with AlexaFluor 488 annexin V and PI for 15 minutes. Samples were immediately analyzed on a Cytotomics FC 500 (Beckman Coulter, Brea, CA) flow cytometer. The percentage of apoptotic/live cells was determined using FlowJo™ as above.

### Western blotting

Protein was isolated from cells after 24-hour drug treatment as described above. Cells were washed with cold PBS and then 150 μL lysis buffer was added to each 10 cm plate. Lysis buffer consisted of 1 mL wash buffer (50 mM Tris pH 7.5, 150 mM NaCl, 0.5% NP-40), 1.0 mM EDTA, 0.2 mM PMSF, and 1X protease inhibitor cocktail (Fisher). For phosphorylation specific analysis,

**Table 1. Antibody information for western blots.**

| Antibody | Dilution | Diluent | Source | Predicted Band Size (kD) |
|---|---|---|---|---|
| Cyclin A2 | 1:200 | 5% NFDM | Cell Signaling Technology (CST) | 55 |
| Cyclin B1 | 1:1000 | 5% BSA | CST | 55 |
| Cyclin D1 | 1:1000 | 5% BSA | CST | 36 |
| Cyclin D3 | 1:2000 | 5% NFDM | CST | 31 |
| Cyclin E1 | 1:1000 | 5% NFDM | CST | 48 |
| CDK1 | 1:1000 | 5% NFDM | CST | 34 |
| CDK1 Thr[14] | 1:1000 | 5% BSA | CST | 34 |
| CDK1 Tyr[15] | 1:1000 | 5% BSA | CST | 34 |
| CDK1 Thr[161] | 1:1000 | 5% BSA | CST | 34 |
| CDK2 | 1:1000 | 5% BSA | CST | 33 |
| CDK4 | 1:1000 | 5% BSA | CST | 30 |
| CDK6 | 1:2000 | 5% NFDM | CST | 36 |
| p18 | 1:1000 | 5% NFDM | CST | 18 |
| p21 | 1:1000 | 1% BSA | Proteintech | 21 |
| p27 | 1:1000 | 5% BSA | CST | 27 |
| HDAC | 1:1000 | 5% BSA | CST | 62 |
| Histone H3.3 | 1:1000 | 5% NFDM | Abcam | 19 |
| β-actin | 1:25,000 | 5% BSA or 5% NFDM | Sigma | 42 |
| NUPR1 | 1:300 | 5% NFDM | Proteintech | 8, 17, 34 |
| GLI1 | 1:1000 | 5% NFDM | CST | 160 |
| APC | 1:500 | 5% NFDM | Abcam | 310 |

1 ml of a phospho-enhancing lysis buffer (20 mM Tris-HCL, 150 mM NaCl, 1% Triton-X, 0.5% NP-40, 50 mM NaF, 1 mM $Na_3VO_4$ and 5mM Sodium Pyrophosphate) was combined with 0.2 mM PMSF, 1X protease inhibitor, and 1X phosphatase inhibitor (Sigma). Cells were then incubated at 4°C for 30 minutes with shaking. The lysate was collected and spun at 15,000 rpm at 4°C for 15 minutes to pellet debris. Protein concentrations were measured using BCA Assay (ThermoFisher) as per manufacturer's instructions and stored at -80°C until further use.

Equal concentrations of protein (10–30 μg depending on antibody) were separated using a 10% SDS-PAGE gel for cell cycle protein analysis or a BioRad Mini-PROTEAN TGX Precast 4–20% gradient gel for fractionated samples or to probe for APC. Protein was then transferred to a 0.45um Immobilon-P PVDF membrane (Millipore). After blocking for 1 hr in 5% nonfat dried milk in TBST, membranes were incubated with primary antibody (Table 1) overnight at 4°C. Membranes were then incubated with a species-specific HRP-conjugated secondary antibody (1:1000 in the same diluent as the primary) for 1 hr at RT. Clarity or Clarity Max ECL reagent and a ChemiDoc MP Imaging System (Bio-Rad) were used to image the blots. β-actin was used as a loading control with a 1hr RT incubation. Analysis of protein levels relative to actin were performed by densitometry using ImageJ software (NIH) [13]. The protein levels of the triplicates were averaged and graphed +/- standard deviation. For the phosphorylation sites, the value graphed was the ratio of the site expression over actin to the total CDK1 protein expression over actin.

## Fractionation

MDA-MB-157 and APC shRNA1 cells were plated in 10 cm dishes, grown to 70% confluency and treated with drugs as above for 24 hours. Cells were then collected using trypsin and

resuspended in PBS. After centrifugation, the pellet was collected and resuspended in 100 uL of cytoplasmic extract buffer (10mM HEPES, 60 mM KCl, 1 mM ETDA, 0.075% (v/v) NP40, 1 mM PMSF, 1mM DTT, adjusted to a pH of 7.6) and incubated on ice for three minutes. The supernatant was removed and placed in a clean tube; this was the cytoplasmic extract (CE). The remaining nuclear pellets were then carefully washed with 100 uL of cytoplasmic extract buffer without detergent (10mM HEPES, 60 mM KCl, 1 mM ETDA, 1 mM PMSF, 1mM DTT, adjusted to a pH of 7.6) and the supernatant was removed and discarded. 50 uL of the nuclear extract buffer (20 mM Tris Cl, 520 mM NaCl, 1,5 mM $MgCl_2$, 25% (v/v) glycerol, 1 mM PMSF, adjusted to pH of 8.0) and 35 uL of 5M NaCl was added to the pellet. An additional 50 uL of nuclear extract buffer was then added and the pellet was resuspended by vortexing. The nuclear extract was incubated on ice for 10 minutes with periodic vortexing. Both the cytoplasmic and nuclear extracts were spun down and the contents were transferred to clean tubes. 20% (v/v) of glycerol was added to the cytoplasmic extracts. Protein quantification was performed, samples were prepared and then analyzed using western blot. HDAC and Histone H3.3 were used as controls for nuclear fractionation.

### RNA sequencing and functional analysis

RNA-sequencing was performed to glean changes occurring in the MDA-MB-157 and APC shRNA1 cells. Control treated and chemotherapy treated cells were assessed through the Genomics Core Facility at the University of Notre Dame. For analysis, raw sequences were trimmed of adapters with Trimmomatic and assessed for quality with FastQC. Trimmed sequences were aligned to the human genome (Homo_sapiens. GRCh38), using corresponding annotations, with TopHat2 using Bowtie 2 [14, 15]. Corresponding alignments were sorted with SAMtools. Read counts were generated with HTSeq-count and were merged with a python script. The merged counts files were normalized using DESeq2 [16]. Functional analysis of the normalized data was done using topGO [17]. topGO provides a convenient means to analyze overrepresentation of GO terms in sample gene lists and gene lists associated with significance scores. The significance of different ontology categories was calculated using different statistics viz. Fisher's exact test, and Kolmogorov-Smirnov test. Genes associated with the GO terms were extracted and identified using the Bioconductor package BioMart [18]. Normalized expression data associated with these genes were used to construct a heatmaps using ggplot2 [19].

### Quantitative analysis by real-time RT-PCR

RNA was isolated from the MDA-MB-157 and APC shRNA1 cells using Tri Reagent (Molecular Research Center, Cincinnati, OH), and cDNA synthesis was performed with iScript (Bio-Rad Laboratories, Hercules, CA). Real-time RT-PCR was performed with Power SYBR Green master mix (Applied Biosystems, Foster City, CA) and 50ng cDNA. Primer sequences are presented in Table 2. The amplification program included: 2 initial steps at 50°C for 2 minutes and 95°C for 10 minutes; 40 cycles of 95°C for 15 seconds and 60°C for 1 minute to allow for denaturation, annealing, and extension; and concluded with generation of a melt curve (CFX

**Table 2. Primer sequences for RT-PCR.**

| | |
|---|---|
| RGS4 –Forward | 5' GCA GGC ATG TGA AGG AGA AAC 3' |
| RGS4 –Reverse | 5' TAT AAG CCC GGC AGC ATA CA 3' |
| LBH–Forward | 5' TCA CTG CCC CGA CTA TCT G 3' |
| LBH–Reverse | 5' GGT TCC ACC ACT ATG GAG G 3' |

Connect 96 thermal cycler, Bio-Rad). Samples were run in duplicate for three independent experiments, with GAPDH as a reference gene to normalize differing levels of expression.

## Statistical analysis

Western blots were analyzed via ImageJ (NIH), and protein quantifications from triplicate runs were averaged and represented using bar graphs with error bars depicting standard deviation unless noted. Real-time RT-PCR reactions were quantified by interpreting the ratio of fluorescence to cycle number to reach threshold (Ct) as relative expression of the gene. Bar graphs of gene expression were then created by averaging biological replicates in Microsoft Excel or GraphPad Prism, and error bars were included to illustrate standard deviation unless noted. Statistical significance was determined using one-way ANOVA with a post-hoc Tukeys t-test.

## Results

### Loss of APC does not alter PTX-induced G2/M arrest in MDA-MB-157 cells

PTX is a well-established spindle poison, working by inducing apoptosis after arresting cells during mitosis. It activates the spindle assembly check point and soon after induces mitotic arrest [6]. APC is also involved in microtubule regulation, but through mechanisms that are not fully known. As expected, the parental MDA-MB-157 cells exhibit apoptosis after 24 hr treatment with PTX; however, the APC shRNA1 cells have a significant decrease in cell death (S2 Fig and [7]). To study the cell cycle profile of the MDA-MB-157 cells and APC shRNA1 cells under control and PTX treatments, flow cytometry was used. Because PTX works by arresting cells during mitosis, we expected an increase in G2/M expression following PTX treatment in cells sensitive to the drug. Nocodazole was used as a positive control because it is known to induce G2/M arrest. Cell cycle analysis indicated that both cell lines exhibited an increase in the G2/M population after PTX treatment (Fig 1). In contrast to our hypothesis, the APC shRNA1 cells arrested in the G2/M phase with PTX, meaning the mechanism behind their survival might include proteins regulating the G2/M checkpoint (Fig 1). This finding led us to further investigate proteins involved in the G2/M phase of the cell cycle.

### Alterations of G2/M checkpoint proteins in APC shRNA1 cells

The G2/M transition is regulated by the protein complex of CDK1 and cyclin B1. Given the increased G2/M arrest in the PTX-treated APC shRNA1 cells and the role of PTX in mitosis, we next investigated the effect of PTX and APC status on the expression of CDK1 and cyclin B1. We found that there was an increase in total CDK1 expression in the untreated and treated APC shRNA1 cells compared to the untreated/treated parental cells (Fig 2), while no change was observed in cyclin B1 expression. We also profiled the inhibitory (Thr[14] and Tyr[15]) and activating (Thr[161]) phosphorylation sites on CDK1, and found no changes in phosphorylation patterns in the APC shRNA1 cells compared to the parental cells. We next assessed whether PTX treatment influenced the expression of cyclin B1 or CDK1, or the phosphorylation status of CDK1. Interestingly, no changes were observed after PTX treatment in either cell line (Fig 2).

### Sub-cellular localization of G2/M checkpoint proteins in APC shRNA1 cells

Altered expression of these checkpoint proteins led us to further explore the CDK1-cyclin B1 complex. The regulation of this complex is essential for proper cell cycle progression. The progression from the G2 phase to the M phase is dependent on the activation and nuclear localization of the CDK1-cyclin B1 complex. The destruction of this complex allows for further

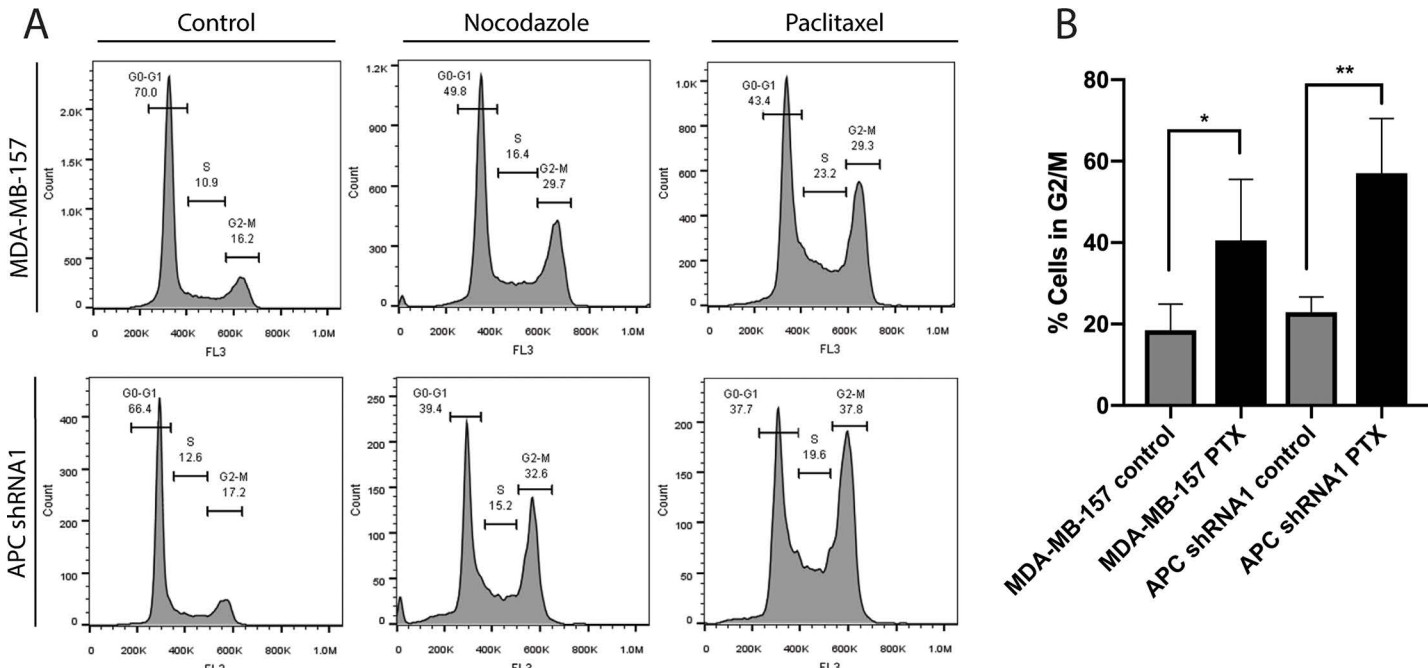

**Fig 1. Analysis of G2/M cell cycle arrest under PTX in MDA-MB-157 & APC shRNA1 cells.** (A) Cell cycle analysis of untreated and PTX treated MDA-MB-157 and APC shRNA1 cells using flow cytometry. Nocodazole was used as a positive control because it is known to induce G2/M arrest. Representative flow cytometry graphs are included. (B) Graphical representation of the percentage of cells in the G2/M phase of the cell cycle in control and PTX treated cells. Data shown are the average of 4 independent experiments, and the STDEV is shown. * p < 0.05, ** p < 0.01 compared to control treated cells of the same genotype.

progression into anaphase. Because the activity of this complex is dependent on its localization in the cell, we examined the expression of CDK1 and cyclin B1 in the cytoplasm and the nucleus using fractionation. Analysis showed that CDK1 was preferentially located in the cytoplasm in the untreated cell lines, while cyclin B1 had no preferential localization (Fig 3). Nocodazole was used as a positive control because it is known to induce G2/M arrest. After treatment with either PTX or nocodazole, no changes were observed in the localization of CDK1.

## Loss of APC or PTX treatment alters expression of cell cycle proteins

Aside from the G2/M checkpoint proteins, CDK1 and cyclin B1, other proteins that regulate cell cycle dynamics, including cyclin/CDK complexes and the CKIs, p27 and p21, have been shown to impact response to taxanes [20–22]. This information led us to interrogate the effect of APC loss on the expression of different cyclins, CDKs, and other cell cycle proteins in the presence or absence of PTX. We found that the G1/S regulator, CDK6, is significantly upregulated in in APC shRNA1 cells (Fig 4A and 4B). Interestingly, p27 expression decreased upon PTX treatment in both cell lines (Fig 4A and 4C). No changes were observed in the other cell cycle proteins investigated (Fig 4).

## RNA sequencing analysis shows cell cycle related gene alterations in APC shRNA1 MDA-MB-157 cells

Given the lack of Wnt pathway activation in the APC shRNA1 MDA-MB-157 cells (S3 Fig), similar to our observations in the MMTV-PyMT;$Apc^{Min/+}$ cells [10], we wanted to assess global gene expression changes to understand the broad cell cycle changes observed. Therefore, we performed RNA sequencing on MDA-MB-157 parent cells and APC shRNA1 cell lines.

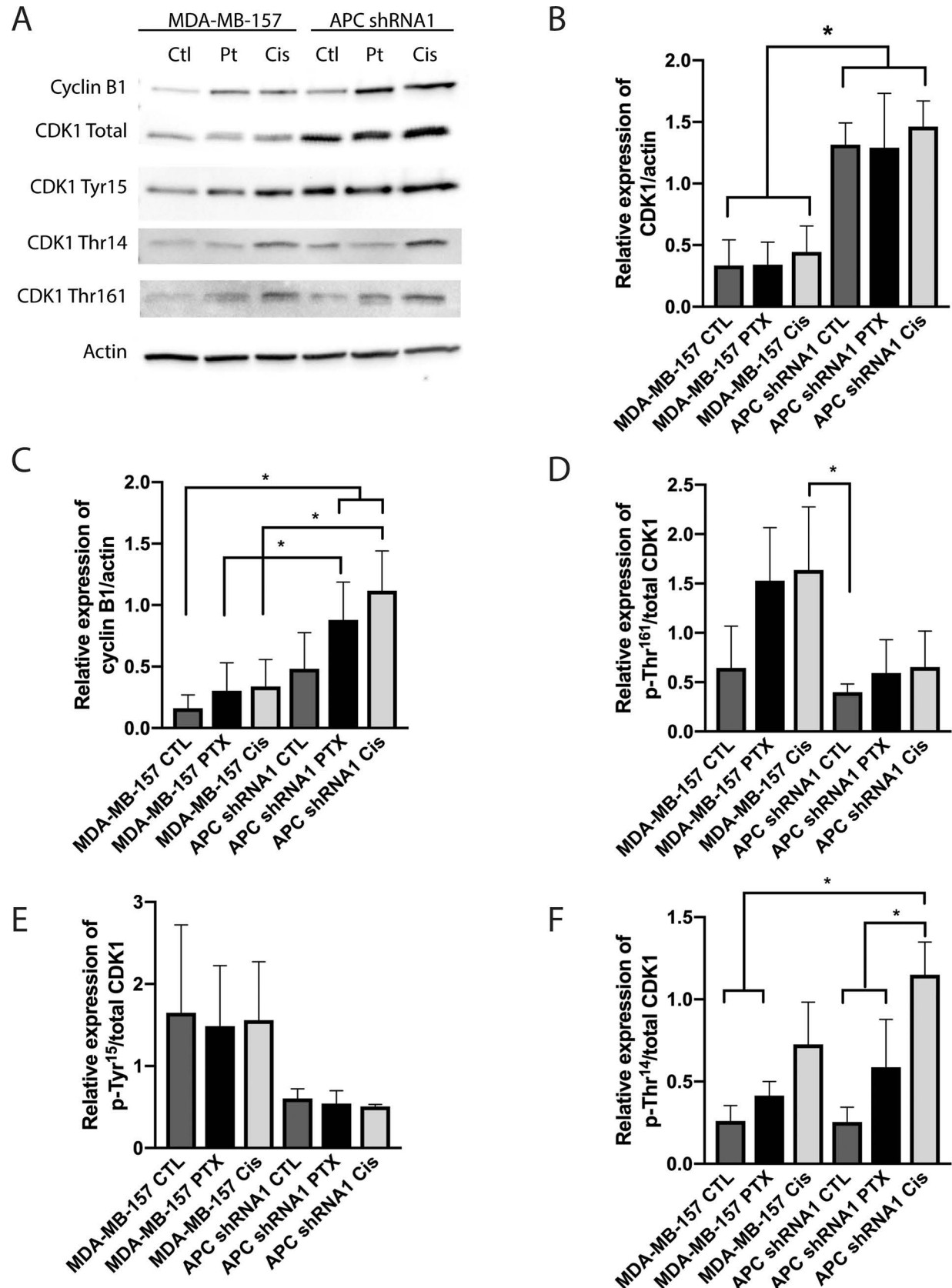

**Fig 2. Western blots of G2/M checkpoint proteins.** (A) Control (CTL) and paclitaxel (PTX) or cisplatin (CIS) treated cell lysates were probed for expression of G2/M checkpoint proteins. Representative western blots are shown for cyclin B1 and CDK1, including the inhibitory (Thr$^{14}$ and Tyr$^{15}$) and activating (Thr$^{161}$) phosphorylation sites on CDK1. (B-F) Bar graph representation of G2/M transition protein quantification in treated and untreated MDA-MB-157 and APC shRNA1 cell lines. Bar graphs show levels of cyclin B1 and CDK1, Thr$^{14}$, Tyr$^{15}$ and Thr$^{161}$ in these treated and untreated cells. For the phosphorylation sites, the value graphed was the ratio of the site expression over actin to the total CDK1 protein expression over actin (n = 3). Analysis indicated an increase in total CDK1 expression in all of the APC shRNA1 cells. * $p < 0.05$ as compared to all of the parental cells. Analysis indicated an increase in phosphorylated CDK1 at Thr$^{14}$ in APC shRNA1 cells upon treatment with cisplatin. * $p < 0.05$ as compared to untreated APC shRNA1 cells, and *** $p < 0.001$ as compared to PTX treated APC shRNA1 cells.

Functional analysis of the transcriptomes of parent cells MDA-MB-157 compared to APC shRNA1 cell lines revealed among others several terms associated with cell cycle and cell division (Table 3). There were over 400 common transcripts differentially expressed between the two cell lines that were associated with cell division and/or cell cycle. A subset of transcripts among these that significantly differentially expressed are shown in the heatmap (Fig 5A). The

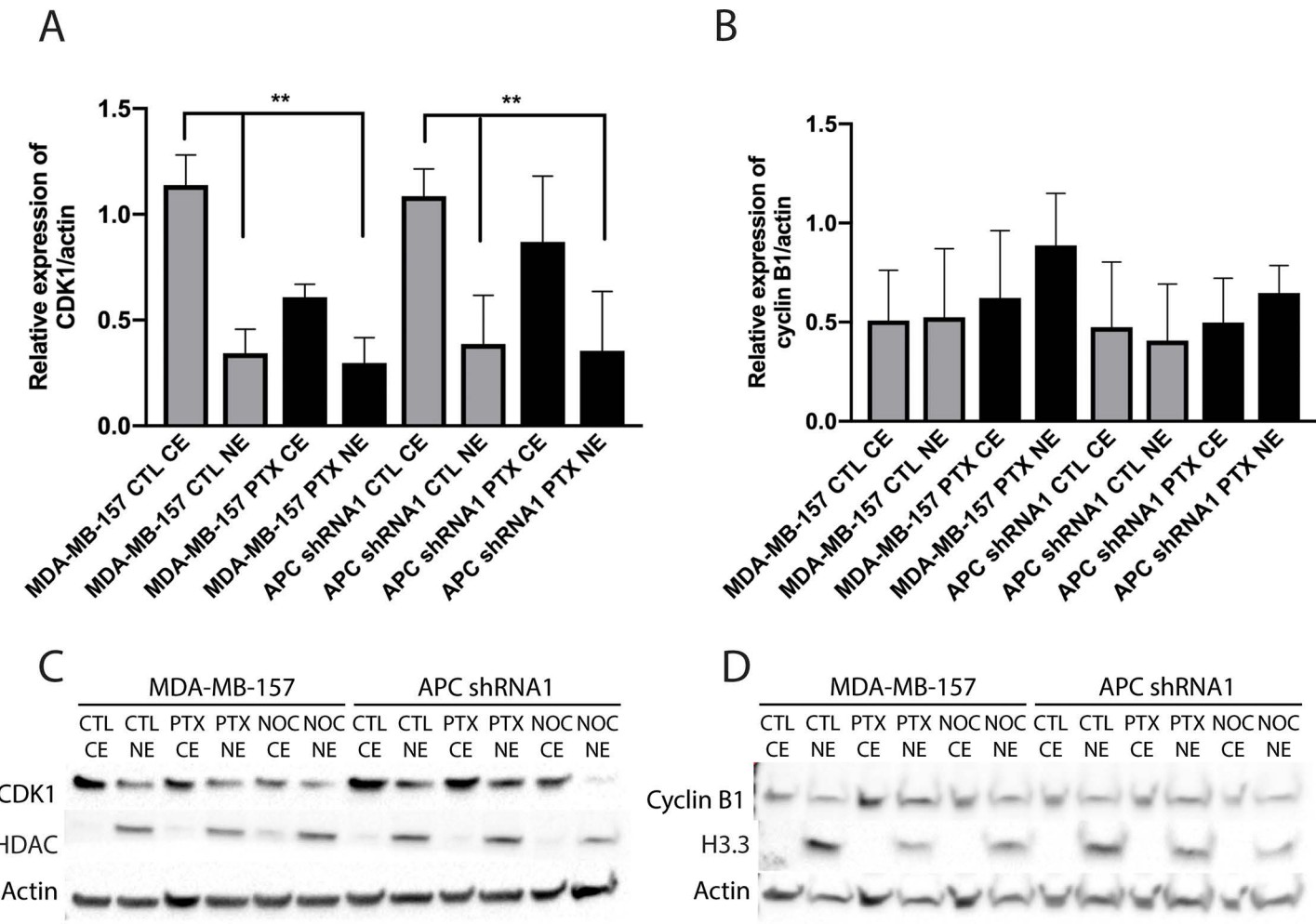

**Fig 3. Western blot analysis of cellular fractions.** Protein lysates from CTL, PTX, or nocodazole (NOC) treated cells were fractionated and then probed with CDK1 (A and C) or cyclin B1 (B and D). HDAC and H3.3 were used as loading controls for the nuclear extract (NE) specifically, while actin was used as a widespread loading control. (A-B) Bar graph representation of protein quantification (n = 3). (C-D) Representative western blots are shown. Analysis indicated that CDK1 is preferentially located in the cytoplasm in the untreated parental and APC shRNA1 cells. Cyclin B1 shows no preferential localization. ** $p < 0.01$ as compared to corresponding cytoplasmic extract (CE). Westerns were run and membranes cut after transfer. For CDK1, the top portion was probed for HDAC (61kD), and the bottom portion was probed for CDK1 (34kD) and actin (42kD). For cyclin B1, the top portion was probed for cyclin B1 (55kD) and actin (42kD), and the bottom was probed for Histone H3.3 (15kD).

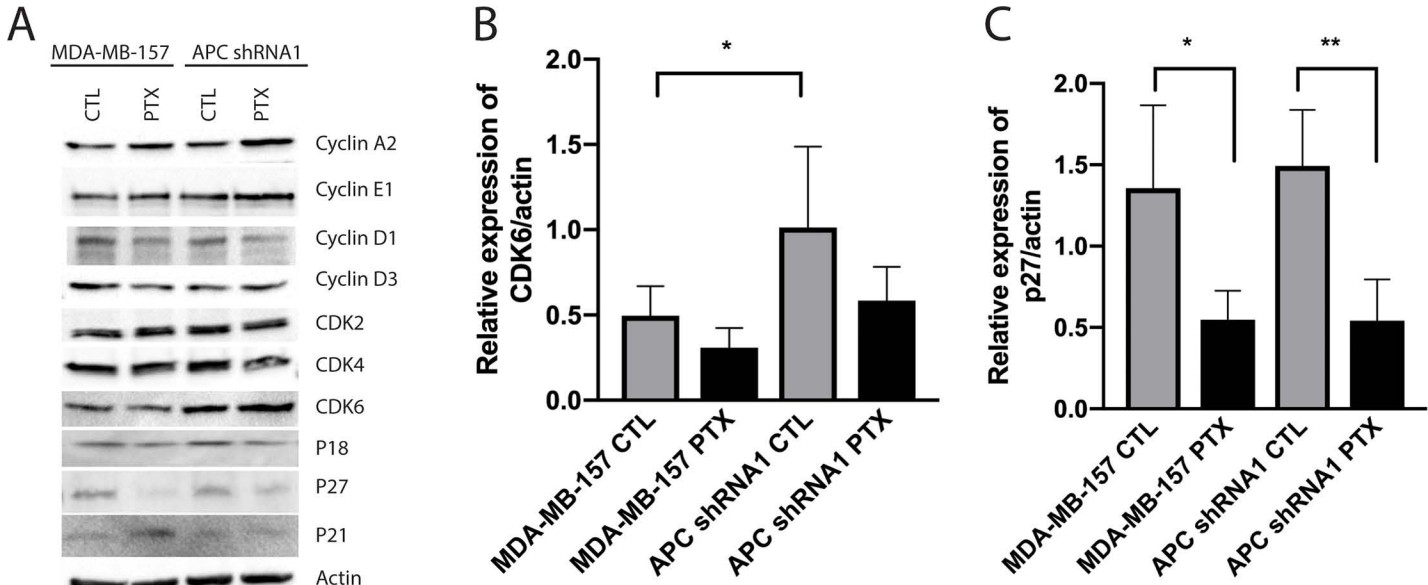

**Fig 4. Western blot analysis of canonical cell cycle proteins.** Untreated or PTX treated lysates were probed for expression of canonical cell cycle proteins. (A) Representative western blots are shown for cyclin A2, cyclin E1, cyclin D1, cyclin D3, CDK2, CDK4, CDK6, p18, p21, and p27. Actin was used as a loading control. (B-C) Bar graph of protein expression quantification in treated and untreated MDA-MB-157 and APC shRNA1 cell lines is shown (n = 3). (B) Analysis indicated that loss of APC resulted in a significant increase in CDK6 expression. * p < 0.05 compared to parental cells. (C) P27 expression decreased upon PTX treatment in both MDA-MB-157 and APC shRNA1 cells. * p < 0.05, ** p < 0.01 compared to control treated cells of the same genotype. No change was seen in the remaining proteins.

entire set is available as a S4 and S5 Figs. Given their functions related to cell cycle phenotypes observed in the APC shRNA1 cells, we confirmed expression of: Glioma-associated oncogene/Zinc finger protein 1 (GLI1); Limb-Bud-Heart (LBH); Nuclear Protein 1 (NUPR1); and Regulator of G-protein Signaling 4 (RGS4). LBH and RGS4 were confirmed using real-time RT-PCR (Fig 5B) and NUPR1 and GLI1 were confirmed via western blot (Fig 5C). In accordance with the RNA-sequencing results, APC shRNA1 cells showed an increase in GLI1, LBH,

**Table 3. Over-represented biological process ontologies of cell cycle/cell division associated with the transcriptome.**

| Sr. No | GO.ID | Term | Annotated | Significant | Expected | Rank in classicKS | P-value classicFisher | P-value classicKS | P-value elimKS |
|---|---|---|---|---|---|---|---|---|---|
| 1 | GO: 0051301 | cell division | 539.00 | 56.00 | 90.72 | 99.00 | 1.00 | 7.50E-13 | 6.70E-10 |
| 2 | GO: 0010972 | negative regulation of G2/M transition o. . . | 79.00 | 6.00 | 13.30 | 273.00 | 0.99 | 1.30E-05 | 0.00014 |
| 3 | GO: 0010971 | positive regulation of G2/M transition o. . . | 26.00 | 0.00 | 4.38 | 433.00 | 1.00 | 0.00025 | 0.00025 |
| 4 | GO: 0070317 | negative regulation of G0 to G1 transiti. . . | 39.00 | 2.00 | 6.56 | 470.00 | 0.99 | 0.0004 | 0.0004 |
| 5 | GO: 0010389 | regulation of G2/M transition of mitotic. . . | 176.00 | 13.00 | 29.62 | 126.00 | 1.00 | 2.90E-10 | 0.00066 |
| 6 | GO: 0000079 | regulation of cyclin-dependent protein s. . . | 91.00 | 8.00 | 15.32 | 540.00 | 0.99 | 0.00069 | 0.00069 |
| 7 | GO: 0044772 | mitotic cell cycle phase transition | 513.00 | 40.00 | 86.34 | 52.00 | 1.00 | 1.20E-17 | 0.00098 |
| 8 | GO: 0007094 | mitotic spindle assembly checkpoint | 32.00 | 0.00 | 5.39 | 608.00 | 1.00 | 0.0011 | 0.0011 |

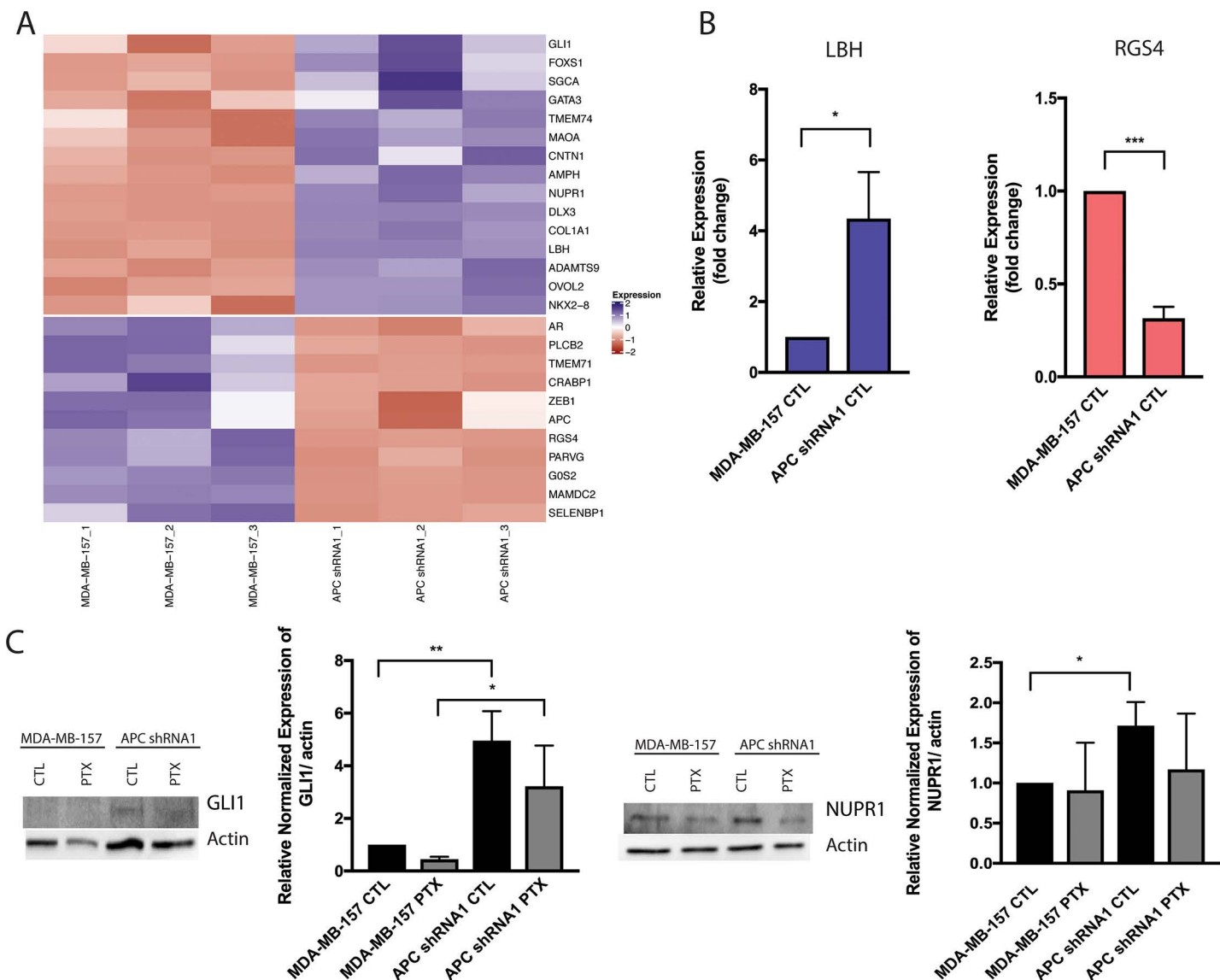

**Fig 5. RNA sequencing and validation of APC shRNA1 cells compared to parental controls.** (A) Hierarchical clustering and heat map of significantly differentially expressed genes associated with the cell cycle. Input data are the normalized expression values. The values in blue are up-regulated and those in red are down-regulated. (B) Real-time RT-PCR analysis for LBH and RGS4 in MDA-MB-157 and APC shRNA1 cells. Data were initially normalized by amount of GAPDH mRNA, and then further normalized to the parental cells. Data (AVG +/- SEM) depict changes in expression between parental and APC shRNA1 cells, with average of parental (n = 3) set to 1 and average of APC shRNA1 representing fold-change. (C) Western blot analysis for GLI1 and NUPR1 in MDA-MB-157 and APC shRNA1 cells with Actin as a loading control. Bar graph representation of protein quantification (n = 3 for GLI1 and n = 4 for NUPR1) in untreated and PTX treated cells. Analysis indicated that loss of APC increased GLI1 and NUPR1 in untreated cells. * $p < 0.05$, ** $p < 0.01$ as compared to parental cells.

and NUPR1 and a decrease in RGS4 expression compared to the MDA-MB-157 parental cell line (Fig 5B and 5C).

## Discussion

Paclitaxel is a β-tubulin-binding chemotherapeutic agent widely used to treat cancers of various types, including TNBC, by disrupting MT dynamics to induce mitotic arrest and apoptosis. Although Paclitaxel's mechanism of action is well characterized, mechanisms of resistance

remain vague, especially in cancers lacking APC. The present study examines the effect of APC status and/or PTX treatment on cell cycle progression and cell cycle protein expression, activation, and localization in MDA-MB-157 cells, specifically at the G2/M interphase. We also identify potential future targets through RNA sequencing that could have implications on cell cycle regulation and the response to PTX.

PI staining and flow cytometry showed that PTX treatment caused an increase of cells in the G2/M phase of the cell cycle in both MDA-MB-157 and APC shRNA1 cells. Similar results have been found by other groups, where Taxol treated, APC-mutant embryonic stem (ES) cells accumulate in the G2/M phase ([23, 24] and reviewed in [25]). These data suggest that APC status does not prevent G2/M arrest after PTX treatment and that the mechanisms required to arrest both the parental and APC shRNA1 cells are functional, but the fate of the cell is altered. The successful G2/M arrest suggests that PTX's mechanism of action is functional, but inherent differences between the parental and APC shRNA1 lead to death in one and survival in the other. This altered cell fate could be due to cell cycle related proteins as discussed here (Figs 2–5) or alterations in proteins that regulate PTX-induced apoptosis, such as the Bcl-2 family of proteins. Many Bcl-2 family members, including Bcl-2, Bax, Bak, and Bcl-xL are known to impact PTX sensitivity in breast cancer [26–28]. Future studies will investigate differences in expression of Bcl-2 family members to explain the resistance to PTX despite the robust accumulation of APC shRNA1 cells in G2/M.

The activation of the mitotic checkpoint involves the protein complex of CDK1-cyclin B1. This complex is involved in cell cycle control, with progression from the G2 to the M-phase being dependent on the activation and nuclear localization of this complex. For an active nuclear complex, CDK1 must be phosphorylated on $Thr^{161}$ and dephosphorylated on $Thr^{14}$ and $Tyr^{15}$. This active complex regulates M-phase entry and exit, and is responsible for activation of the anaphase promoting complex (APC/C) [29]. Further progression into anaphase is dependent on destruction of the active CDK1-cyclin B1 complex, which is mediated by APC/C [30]. Therefore, we analyzed the expression, activation, and subcellular localization of these important G2/M checkpoint proteins. Examination of protein expression by western blot showed upregulated levels of CDK1 in APC shRNA1 cells compared to the parental cell line (Fig 2). We also examined activation of CDK1 by profiling phosphorylation sites on CDK1 including the inhibitory ($Thr^{14}$ and $Tyr^{15}$) and activating ($Thr^{161}$) phosphorylation sites, finding no significant change in CDK1 activation status between parental and APC shRNA1 cell lines (Fig 2). However, a possible explanation to the increased levels of CDK1 without activating phosphorylation could be that the APC shRNA1 cells are promoting mitotic slippage by slowly degrading levels of cyclin B1 below threshold, resulting in mitotic exit, despite the spindle assembly checkpoint being left unsatisfied and the CDK1-cyclin B1 complex inactive [31–37]. Moreover, connections between the CDK1-cyclin B1 complex and members of the apoptotic pathway provide a possible link between promoting mitosis and regulating cell death [38–40]. Future studies should include investigating APC/C, cdc20, and SAC activity to determine if slippage is in fact occurring and which component is responsible for it. Furthermore, evaluating pro- and anti-apoptotic proteins and their interactions with regulators of the cell cycle could identify mechanism of protection from PTX-induced apoptosis.

Examination of subcellular protein localization by fractionation indicated that CDK1 is preferentially located in the cytoplasm in the untreated cell lines, while cyclin B1 has no preferential localization (Fig 3). This also suggests that CDK1 may have a cytoplasmic function besides binding with cyclin B1 and translocating to the nucleus, perhaps by interacting with apoptotic family members [38–40]. Targeting mitotic control, either by selective inhibition of CDK1 or aberrant activation of CDK1 via Wee1 inhibition, has been successful in increasing the efficacy of PTX in breast and ovarian cancers [41, 42]. Future studies will examine the

effect of genetic and chemical CDK1 inhibition on PTX sensitivity in MDA-MB-157 and APC shRNA1 cells.

Since other cyclins and CDKs have been shown to mediate PTX sensitivity, as well as CDK1 activity, we also examined expression of other cell cycle proteins [20–22]. Protein expression by western blot indicated that APC shRNA1 cells have significantly increased CDK6, which has been frequently observed in different cancer types [37]. Therefore, future work in the lab will examine the effect of CDK6 inhibition on PTX sensitivity. Since the expression and localization of p27 has been linked to tumorigenic and chemoresistant phenotypes [43, 44], and PTX treatment decreased p27 expression in both cell lines (Fig 4), future studies will investigate how the localization of p27 impacts response to PTX.

RNA sequencing revealed 403 common transcripts between the two APC shRNA cell lines, including those involved in expression of cell cycle proteins (S4 and S5 Figs and Fig 5A). In accordance with the RNA sequencing results, RT-PCR validation showed an increase in LBH and a decrease in RGS4 expression (Fig 5B) and Western Blot validation demonstrated an increase in GLI1 and NUPR1 (Fig 5C) in APC shRNA1 cells compared to the parental cells. NUPR1 is a transcriptional regulator that is upregulated in response to cell stress and therefore is involved in many pathways including regulating the cell cycle, apoptosis, and DNA repair. It has been shown that NUPR1 is associated with poor prognosis as well as chemoresistance in breast cancer [45–47]. An increase in NUPR1 in APC shRNA1 cells could be a possible mechanism for evading apoptosis induced by chemotherapy or arresting in G2/M. GLI1 is an effector protein of the Hedgehog (Hh) pathway, involved in cell development and differentiation. Increased expression of GLI1 has been shown to be associated with metastasis, increased proliferation, and the enrichment of cancer stem cells [48–50]. Increased transcription of cell proliferation and cell survival genes, such as cyclin D1 and Bcl-2, has been linked to the active Hh pathway [51]. Therefore, increased GLI1 could link cell cycle alteration in the APC shRNA1 cells with increased Hh pathway activity. Our lab has previously shown that APC loss-of-function results in increased tumor initiating cells (TICs) [7]. Given the role of TICs in developing drug resistance (reviewed in [52]), the increased GLI1 expression in APC shRNA1 cells could indicate an increase in TICs, leading to PTX resistance. Aside from being a key regulator in the Wnt/β-catenin pathway, LBH can regulate aspects of the cell cycle and is highly expressed in aggressive basal subtype breast cancers [53, 54]. LBH deficiency has been shown to arrest cells S phase by altering cell cycle protein expression and DNA damage repair pathways [54]. Therefore, the increased LBH (Fig 5) could be contributing to the cell cycle alterations observed in APC shRNA1 cells. RGS4 is a GTPase activating protein that forces G proteins into their inactive form. Studies have shown loss of RGS4 in multiple cancer types, including non-small cell lung cancer [55], melanoma [56], and breast cancer [57, 58]. RGS4 expression has been shown to decrease in response to cell stress [59], and lower cellular levels have been associated with increased proliferation and involvement with cell cycle arrest in the G2/M phase [60]. Future studies could investigate whether APC shRNA1 cells have increased proteasomal degradation of RGS4 to activate GTPases involved in anti-apoptotic pathways. Together, these genes may be intermediate molecular markers between APC status and response to PTX. Future studies will explore the impact of these gene expression changes on PTX resistance.

Our lab has found that APC mediates chemotherapeutic response in several TNBCs, including the response of MDA-MB-157 and APC shRNA1 cells to PTX. The mechanisms by which PTX and APC bind β-tubulin to effect the G2 and M phases are well characterized, but APC-mediated mechanisms of PTX resistance remain unexplored. An arrest in G2/M following PTX treatment in APC knockdown cells indicates the interaction between APC and tubulin might not be as relevant to deciphering resistance mechanisms, since the PTX mechanism of action is still taking place. Due

to the effects of PTX treatment on G2/M arrest, exploration of the kinases and cell cycle proteins associated with the G2/M transition may provide insight to the mechanism of PTX resistance. Examining cell cycle protein levels over a time course post-PTX treatment, targeting the kinases and genes involved in cell cycle progression indicated by RNA seq to be altered by APC status, and interrogating how APC status effects levels and activity of apoptotic pathway members are of utmost importance in elucidating the role of APC-mediated PTX resistance in TNBC.

## Conclusions

Loss of the APC tumor suppressor in human TNBC cells alters the response to taxane treatment, likely through cell cycle mediated changes. Regulation of CDK1, p27, and CDK6 at the protein level is dependent on APC expression. Combined, this suggests that APC status may be useful as a marker for taxane-resistant TNBC, and that targeting downstream of APC could be a novel therapeutic approach for TNBC.

## Supporting information

**S1 Fig. Western blot analysis of APC expression in parental and APC shRNA1 cells.**
(A) Untreated cell lysates were probed for APC expression and representative western blot is shown. (B) Bar graph displays average (n = 3) expression of APC relative to actin in MDA-MB-157 and APC shRNA1 cells.
(PDF)

**S2 Fig. Annexin V and PI staining of parental and APC shRNA1 cells.** MDA-MB-157 and APC shRNA1 cells were treated with PTX or DMSO control and stained for annexin V and PI. (A) Representative histograms of the apoptotic population in Q2 (late apoptosis) and Q3 (early apoptosis). (B) Quantification of the combined apoptotic population (n = 3). * p < 0.05 comparing PTX to DMSO treated parental MDA-MB-157 cells. No difference was observed after PTX treatment in the APC shRNA1 cells.
(PDF)

**S3 Fig. β-catenin/TCF reporter assays.** β-catenin/TCF reporter assays showed minimal basal Wnt/β-catenin pathway activation in the APC shRNA1 cells compared to the parental MDA-MB-157. SW480 cells were used as a positive control. The data are shown as a ratio of normalized TOP-Flash values. **** p < 0.0001 compared to SW480 control cells.
(PDF)

**S4 Fig. Heat map and gene description for genes down-regulated in the APC shRNA1 cells.**
(A) Hierarchical clustering and heat map of the genes associated with the over-represented biological process ontologies of cell cycle/cell division. Input data are the normalized expression values. The values in blue are upregulated and those in red are down-regulated. (B) The description of each gene in the clusters are available from the associated excel file.
(PDF)

**S5 Fig. Heat map and gene description for genes up-regulated in the APC shRNA1 cells.**
(A) Hierarchical clustering and heat map of the genes associated with the over-represented biological process ontologies of cell cycle/cell division. Input data are the normalized expression values. The values in blue are upregulated and those in red are down-regulated. (B) The description of each gene in the clusters are available from the associated excel file.
(PDF)

**S1 Raw images.**
(PDF)

## Acknowledgments

The authors thank Abby Wasierski for technical assistance, and Casey Stefanski for reviewing the manuscript. We thank Dr. Charles Tessier at the IUSM-SB Flow and Imaging Core Facility for assistance in data collection and analysis.

## Author contributions

**Conceptualization:** Jenifer R. Prosperi.

**Formal analysis:** Sara M. Maloney, T. Murlidharan Nair.

**Funding acquisition:** Emily M. Astarita, Bronwyn J. Berkeley, Jenifer R. Prosperi.

**Investigation:** Emily M. Astarita, Sara M. Maloney, Camden A. Spring, Bronwyn J. Berkeley, Monica K. VanKlompenberg, Jenifer R. Prosperi.

**Methodology:** Emily M. Astarita, Sara M. Maloney, Camden A. Spring, Monica K. VanKlompenberg, Jenifer R. Prosperi.

**Writing – original draft:** Emily M. Astarita, Sara M. Maloney.

**Writing – review & editing:** Camden A. Spring, Jenifer R. Prosperi.

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
