## [Decision Letter · Decision Letter 0]

29 Mar 2021

PONE-D-21-06112

Adenomatous Polyposis Coli loss controls cell cycle regulators and response to paclitaxel

PLOS ONE

Dear Dr. Prosperi,

Thank you for submitting your manuscript to PLOS ONE. After careful consideration, we feel that it has merit but does not fully meet PLOS ONE’s publication criteria as it currently stands. Therefore, we invite you to submit a revised version of the manuscript that addresses the points raised during the review process.

We look forward to receiving your revised manuscript.

Kind regards,

Sumitra Deb, PhD

Academic Editor

PLOS ONE

Journal Requirements:

Reviewers' comments:

Reviewer's Responses to Questions

**Comments to the Author**

1. Is the manuscript technically sound, and do the data support the conclusions?

Reviewer #1: No

Reviewer #2: No

Reviewer #3: Partly

2. Has the statistical analysis been performed appropriately and rigorously?

Reviewer #1: No

Reviewer #2: Yes

Reviewer #3: Yes

3. Have the authors made all data underlying the findings in their manuscript fully available?

Reviewer #1: Yes

Reviewer #2: Yes

Reviewer #3: Yes

4. Is the manuscript presented in an intelligible fashion and written in standard English?

Reviewer #1: No

Reviewer #2: Yes

Reviewer #3: Yes

5. Review Comments to the Author

Reviewer #1: Most of the data are reported in this paper on the same cell line as breast:

APC selectively mediates response to chemotherapeutic agents in breast cancer.

VanKlompenberg MK, Bedalov CO, Soto KF, Prosperi JR.BMC Cancer. 2016 Nov 28;16(1):921. doi: 10.1186/s12885-016-2950-5.

I think that the paper is not suitable for the publication in the present form, most of the experiments are not convincing and regarding the statistical analysis I have several doubts about the tests used p value with **** does not exist.

Reviewer #2: The authors of this manuscript draw the attention of the readers to tumor suppressor, ‘Adenomatous Polyposis Coli (APC)’ and that the loss of APC in TNBC influences the response to Paclitaxel treatment via cell cycle regulators.

The study however lacks few major experiments to prove this concept.

1. The study has considered single TNBC cell line MD-MB-157 to study the loss of APC in the resistance to Paclitaxel. Experiments should be performed in atleast two or more APC modified TNBC cell lines. Furthermore, cell proliferation assay and annexin staining should be shown in the manuscript in both parental and APC modified cell line with and without Paclitaxel treatment.

2. The blot for CDK1 Thr161, an important marker of CDK1 activation is not visible at all. Also, please indicate the molecular size of the bands.

3. Relative expression changes of CDK6 and p27 calculated in Figure 4 do not match visibly with that of the representative blot. Expression of p27 in the APC shRNA CTL lysates appear decreased compared to parental untreated CTL lysates. The authors should include an explanation for the decrease levels of CDK6 and p27 levels observed on Paclitaxel treatment in the APC modified lines.

4. Again, blot of Paclitaxel treated parental lines (GLI1 and NUPR1) from Figure 5C does not match with the relative expression levels quantified.

5. The author should provide an explanation for no significant changes seen in the GLI1 and NUPR1 levels of Paclitaxel treated parental lines?

6. Knockdown or over expression studies of cell cycle regulators identified via RNA seq need to be majorly performed in APC modified or parental lines proving loss of APC and the likely mechanism of resistance in response to Paclitaxel.

Reviewer #3: The manuscript is well written and easy to follow

1.Major concerns:

1.1 All experiments in this work are performed on MDA-MB-157 and its clonal derivative, APC shRNA cells. This raises a concern if the reported effects of Apc loss are unique to this cell line. Is MDA-MB-157 a representative cell line of all TNBCs? If yes, then sufficient proof by providing references from earlier works should be provided. If no, a few other TNBC cell lines need to be included. Else, the title is not justified. One suggestion to take care of this concern, is to include at least two other TNBC cell lines and a “normal” breast cell line such as MCF-10A and compare the expression profiles of the reported genes by using WB/qPCR following siRNA APC knockdown.

1.2 The authors make an important observation (supplementary figure S2 and line 301) that Apc loss does not activate the Wnt beta-catenin dependent pathway in MDA-MB-157 cells and MMTV-PyMT; ApcMin/+ cells (from their earlier study) and highlight that Apc has beta-catenin independent functions. Is this phenomenon unique to these two cell lines? Or is it also observed in other TNBC cell lines? A TopFlash reporter assay or detection of active beta-Catenin by a western blot in a few more TNBC cell lines is suggested.

2. Minor corrections:

2.1 The reference (7) in line 302 should be corresponding to your earlier article, “Apc Mutation Enhances PyMT-Induced Mammary Tumorigenesis” (fig S3) and not “APC selectively mediates response to chemotherapeutic agents in breast cancer”

2.2 A brief description of the TopFlash reporter assay could be included in the Methods section.

6. PLOS authors have the option to publish the peer review history of their article (what does this mean? ). If published, this will include your full peer review and any attached files.

Reviewer #1: No

Reviewer #2: No

Reviewer #3: No

---

## [Decision Letter · Decision Letter 1]

16 Jul 2021

PONE-D-21-06112R1

Adenomatous Polyposis Coli loss controls cell cycle regulators and response to paclitaxel

PLOS ONE

Dear Dr. Prosperi,

Thank you for submitting your manuscript to PLOS ONE. After careful consideration, we feel that it has merit but does not fully meet PLOS ONE’s publication criteria as it currently stands. Therefore, we invite you to submit a revised version of the manuscript that addresses the points raised during the review process.

We look forward to receiving your revised manuscript.

Kind regards,

Sumitra Deb, PhD

Academic Editor

PLOS ONE

Journal Requirements:

Reviewers' comments:

Reviewer's Responses to Questions

**Comments to the Author**

1. If the authors have adequately addressed your comments raised in a previous round of review and you feel that this manuscript is now acceptable for publication, you may indicate that here to bypass the “Comments to the Author” section, enter your conflict of interest statement in the “Confidential to Editor” section, and submit your "Accept" recommendation.

Reviewer #2: All comments have been addressed

Reviewer #3: (No Response)

2. Is the manuscript technically sound, and do the data support the conclusions?

Reviewer #2: Yes

Reviewer #3: Partly

3. Has the statistical analysis been performed appropriately and rigorously?

Reviewer #2: Yes

Reviewer #3: I Don't Know

4. Have the authors made all data underlying the findings in their manuscript fully available?

Reviewer #2: Yes

Reviewer #3: No

5. Is the manuscript presented in an intelligible fashion and written in standard English?

Reviewer #2: Yes

Reviewer #3: Yes

6. Review Comments to the Author

Reviewer #2: The authors of the manuscript have mostly addressed the reviewer's comments. I believe the authors would take into consideration the reviewers suggestions for further studies and publications.

Reviewer #3: 1. The title of the manuscript is still not justified since the study has been conducted on a single cell line. Either, the authors are encouraged to use multiple cell lines to justify the current title or include MDA-MB-157 in the title itself.

2. The heading of paragraph "APC shRNA1 cells have increased G2/M arrest after PTX treatment compared to MDA-MB-157" (lines 222 and 223) is not justified because it contradicts with the results from the cell cycle analysis experiments (Fig 1) and your own interpretations (please see lines 30 to 32 and again lines 234 to 236). Therefore, authors are encouraged to either perform more cell cycle experiments to get better standard deviation values to justify their claim or rephrase the heading.

3. If TopFlash reporter assays were performed in other cell lines (MDA-MB-231 and SUM159) in the current study then those results need to be included in Fig S3 or appropriately cited if they are from earlier work.

7. PLOS authors have the option to publish the peer review history of their article (what does this mean? ). If published, this will include your full peer review and any attached files.

Reviewer #2: No

Reviewer #3: No

---

## [Author Response · Author response to Decision Letter 1]

21 Jul 2021

Editorial Review

Please review your reference list to ensure that it is complete and correct. If you have cited papers that have been retracted, please include the rationale for doing so in the manuscript text, or remove these references and replace them with relevant current references. Any changes to the reference list should be mentioned in the rebuttal letter that accompanies your revised manuscript. If you need to cite a retracted article, indicate the article’s retracted status in the References list and also include a citation and full reference for the retraction notice. The references have been confirmed and no changes were necessary.

Reviewer #2:

The authors of the manuscript have mostly addressed the reviewer's comments. I believe the authors would take into consideration the reviewers suggestions for further studies and publications. We appreciate the comments.

Reviewer #3:

1. The title of the manuscript is still not justified since the study has been conducted on a single cell line. Either, the authors are encouraged to use multiple cell lines to justify the current title or include MDA-MB-157 in the title itself. The title has been amended to reflect the MDA-MB-157 cells being used in this study.

2. The heading of paragraph "APC shRNA1 cells have increased G2/M arrest after PTX treatment compared to MDA-MB-157" (lines 222 and 223) is not justified because it contradicts with the results from the cell cycle analysis experiments (Fig 1) and your own interpretations (please see lines 30 to 32 and again lines 234 to 236). Therefore, authors are encouraged to either perform more cell cycle experiments to get better standard deviation values to justify their claim or rephrase the heading. Thank you for the observation. The heading has been revised to indicate that APC status (in the MDA-MB-157 cells) doesn’t impact the ability of PTX to induce G2/M arrest.

3. If TopFlash reporter assays were performed in other cell lines (MDA-MB-231 and SUM159) in the current study then those results need to be included in Fig S3 or appropriately cited if they are from earlier work. Thank you for acknowledging the important work done here. Given that this paper is focused on the MDA-MB-157 human metaplastic breast cancer cell line (reflected in the title change also), we do not feel that it’s fitting to include the data on reporter activity in other cell lines. There is no discussion in the paper on this lack of Wnt activity being universal (or in other TNBC cell lines), so we are following the PLOS data policy of sharing all data.

---

## [Editor Report · Decision Letter 2]

23 Jul 2021

Adenomatous Polyposis Coli loss controls cell cycle regulators and response to paclitaxel in MDA-MB-157 metaplastic breast cancer cells

PONE-D-21-06112R2

Dear Dr. Prosperi,

We’re pleased to inform you that your manuscript has been judged scientifically suitable for publication and will be formally accepted for publication once it meets all outstanding technical requirements.

An invoice for payment will follow shortly after the formal acceptance. To ensure an efficient process, please log into Editorial Manager at http://www.editorialmanager.com/pone/ , click the 'Update My Information' link at the top of the page, and double check that your user information is up-to-date. If you have any billing related questions, please contact our Author Billing department directly at authorbilling@plos.org.

Kind regards,

Sumitra Deb, PhD

Academic Editor

PLOS ONE
---

## [Editor Report · Acceptance letter]

30 Jul 2021

PONE-D-21-06112R2

Adenomatous Polyposis Coli loss controls cell cycle regulators and response to paclitaxel in MDA-MB-157 metaplastic breast cancer cells

Dear Dr. Prosperi:

I'm pleased to inform you that your manuscript has been deemed suitable for publication in PLOS ONE. Congratulations! Your manuscript is now with our production department.

Kind regards,

on behalf of

Dr. Sumitra Deb

Academic Editor

PLOS ONE